# Spin-orbital Yu-Shiba-Rusinov states in single Kondo molecular magnet

Hui-Nan Xia[1], Emi Minamitani [2,9], Rok Žitko [3,4], Zhen-Yu Liu[1], Xin Liao[1], Min Cai[1], Zi-Heng Ling[1], Wen-Hao Zhang [1], Svetlana Klyatskaya[5], Mario Ruben [5,6,7] & Ying-Shuang Fu [1,8] ✉

Studies of single-spin objects are essential for designing emergent quantum states. We investigate a molecular magnet $Tb_2Pc_3$ interacting with a superconducting Pb(111) substrate, which hosts unprecedented Yu-Shiba-Rusinov (YSR) subgap states, dubbed spin-orbital YSR states. Upon adsorption of the molecule on Pb, the degeneracy of its lowest unoccupied molecular orbitals (LUMO) is lifted, and the lower LUMO forms a radical spin via charge transfer. This leads to Kondo screening and subgap states. Intriguingly, the YSR states display two pairs of resonances with clearly distinct behavior. The energy of the inner pair exhibits prominent inter and intra molecular variation, and it strongly depends on the tip height. The outer pair, however, shifts only slightly. As is unveiled through theoretical calculations, the two pairs of YSR states originate from the ligand spin and charge-fluctuating higher LUMO, coexisting in a single molecule, but only weakly coupled presumably due to different spatial distribution. Our work paves the way for understanding complex many-body excitations and constructing molecule-based topological superconductivity.

The exchange coupling between magnetic impurities and Bogoliubov quasiparticles of a superconductor leads to low-energy bound states inside the superconducting gap known as Yu-Shiba-Rusinov (YSR) states[1–3]. The YSR states serve as a sensitive probe for studying intricate interactions. This capability is made possible by their sharp peak widths because of their intragap (bound state) nature that prolongs their lifetime[4]. A single spin of magnetic impurity conventionally creates a single pair of YSR states. YSR multiplet states with several pairs may also occur reflecting different interaction effects, whose origin ranges from angular momenta[5], magnetic anisotropy[6], magnetic exchange interactions[7–9], to the orbital structure of the local magnetic moment[4,10].

In addition to unraveling the intricate interactions, the YSR states can serve as building blocks for designing exotic emergent states. In YSR chains or lattices, their hybridization may form YSR bands of topological character, which support non-Abelian Majorana modes at the boundary[11–15]. To date, YSR-type topological superconductors have been constructed with magnetic atoms[16–19]. Molecular magnets hold special virtues over magnetic atoms, because of their structural stability, self-assembly capability, and chemical versatility[20]. However, their spins are usually packaged inside spin-inactive molecular skeletons, preventing their coupling for building molecule-based topological superconductors[21]. In contrast, molecules with exposed ligand

[1]School of Physics and Wuhan National High Magnetic Field Center, Huazhong University of Science and Technology, Wuhan 430074, China. [2]Institute for Molecular Science, Okazaki 444-8585, Japan. [3]Jožef Stefan Institute, Jamova 39, SI-1000 Ljubljana, Slovenia. [4]Faculty of Mathematics and Physics, University of Ljubljana, Jadranska 19, SI-1000 Ljubljana, Slovenia. [5]Institute of Nanotechnology (INT), Karlsruhe Institute of Technology (KIT), Hermann-von-Helmholtz-Platz 1, 76344 Eggenstein-Leopoldshafen, Germany. [6]Institute for Quantum Materials and Technologies (IQMT), Karlsruhe Institute of Technology (KIT), Hermann-von-Helmholtz-Platz 1, 76344 Eggenstein-Leopoldshafen, Germany. [7]Centre Européen de Sciences Quantiques (CESQ), Institut de Science et d'Ingénierie Supramoléculaires (ISIS), 8 allée Gaspard Monge, BP 70028, 67083 Strasbourg Cedex, France. [8]Hubei Key Laboratory of Gravitation and Quantum Physics, Huazhong University of Science and Technology, Wuhan 430074, China. [9]Present address: The Institute for Scientific and Industrial Research, Osaka University, Mihogaoka 8-1, Ibaraki, Osaka 567-0047, Japan. ✉e-mail: yfu@hust.edu.cn

spins are readily coupled and thereby represent excellent candidates in that regard[22,23]. This calls for a thorough investigation of the properties of single molecular YSR states with extended ligand spins. More importantly, the ligand spins are hosted by molecular orbitals with rich textures, potentially giving rise to fundamentally new mechanism of spin-orbital entangled YSR excitations.

Here, we report an unprecedented type of YSR state, due to the interplay between a local moment and another valence-fluctuating orbital, in ligand spins of $Tb_2Pc_3$ molecules, phthalocyanine-based triple-decker dinuclear sandwich-type terbium complex (see Fig. 1a for its molecular structure), on superconducting Pb(111). Spectroscopic imaging scanning tunneling microscopy (SI-STM) reveals two-pair YSR states delocalized over the entire $Tb_2Pc_3$. The inner YSR pair shows prominent energy evolution with changing molecule-substrate interaction, while the outer YSR pair remains nearly unshifted. First principles calculations unveil that the substrate adsorption of $Tb_2Pc_3$ lifts the two-fold degeneracy of its lowest unoccupied molecular orbital (LUMO) and the higher split LUMO possesses charge fluctuations. This leads to a spin-orbital correlated system, inducing two-pair YSR states, dubbed spin-orbital YSR states, that are reproduced in numerical renormalization group (NRG) simulation. Our study identifies an unexplored mechanism of inducing multiple intragap excitations, paving the way for studying coupled many-body YSR states.

## Results

The measurements were performed with a custom-made Unisoku STM system (1300) at 0.4 K. Double-decker $TbPc_2$ molecules were deposited on Pb(111) surface at room temperature. $Tb_2Pc_3$ molecule contains two Tb ions sandwiched by three Pc ligands that mutually stack by 45° (Fig. 1a). $TbPc_2$ shares a similar structure as $Tb_2Pc_3$ but with one TbPc unit missing (Fig. 1a). $TbPc_2$ molecules self-assemble into square lattice upon surface adsorption with several sparsely imbedded brighter $Tb_2Pc_3$ molecules (Fig. 1b) (Supplementary Note 1). The $Tb_2Pc_3$ molecules are formed via a chemical reaction from $TbPc_2$ precursors that decompose into TbPc and Pc at a high temperature of 650–700 K in the crucible, following a reaction pathway: $4TbPc_2 \rightarrow Tb_2Pc_3 + TbPc_2 +$

$TbPc+2Pc^{24}$. The formation of $Tb_2Pc_3$ molecules can be evidenced from its manipulation as a single entity by the STM tip (Supplementary Fig. S1a–c), as well as the existence of single decker species Pc and TbPc on the surface, whose estimated statistical ratio 1.8 is close to the value of 2 expected from the reaction pathway (Supplementary Fig. S1d, e).

Isolated $TbPc_2$ has a π radical spin delocalized over the two Pc ligands. The f-spin of Tb possesses high magnetic anisotropy exerted from the ligand field, but is too localized[25,26] and plays no role in our experiments. Isolated $Tb_2Pc_3$ has no radical spin due to the complete pair of its π system by an additional Pc deck[26]. The spin states of the molecules change upon substrate absorption, as determined from their low energy spectra described in the following. First, an out-of-plane field of 2 T is applied to quench the superconductivity of Pb. Tunneling spectra of $TbPc_2$ show featureless constant conductance at low bias. $Tb_2Pc_3$ molecules are classified into two types from distinct spectroscopic features (Fig. 1c). The type 1 $Tb_2Pc_3$ has a sharp Kondo resonance around the Fermi level $E_F$, as is directly evidenced from its expected Zeeman splitting (Supplementary Fig. S3)[27]. The type 2 $Tb_2Pc_3$, however, shows no Kondo resonance.

At zero magnetic field, both the $TbPc_2$ and the type 2 $Tb_2Pc_3$ display the same superconducting gaps as the Pb substrate (Fig. 1d). This indicates that the ligand spin of $TbPc_2$ is quenched upon adsorption[28], and that the type 2 $Tb_2Pc_3$ remains nonmagnetic. On the contrary, the type 1 $Tb_2Pc_3$ exhibits two pairs of sharp YSR excitations within the superconducting gap that are symmetric with respect to $E_F$ (Fig. 1d). These subgap states are the focus of this work. The emergence of the YSR states indicates the activation of the ligand spin induced by substrate charge transfer. The nonmagnetic state in type 2 $Tb_2Pc_3$ suggests, instead, a negligible charge transfer.

The different charge transfer of $Tb_2Pc_3$ can be unraveled from the spectroscopic mapping of the $TbPc_2$ film at 0.5 V, showing a moiré superstructure (Supplementary Fig. S4). $TbPc_2$ at different moiré positions exhibit evident energy shift of the molecular orbitals (Fig. 1e). The type 1 $Tb_2Pc_3$ occupies the moiré region with lower conductance in Fig. 1f, whose LUMO is closer to $E_F$ than that of the type 2 $Tb_2Pc_3$ by

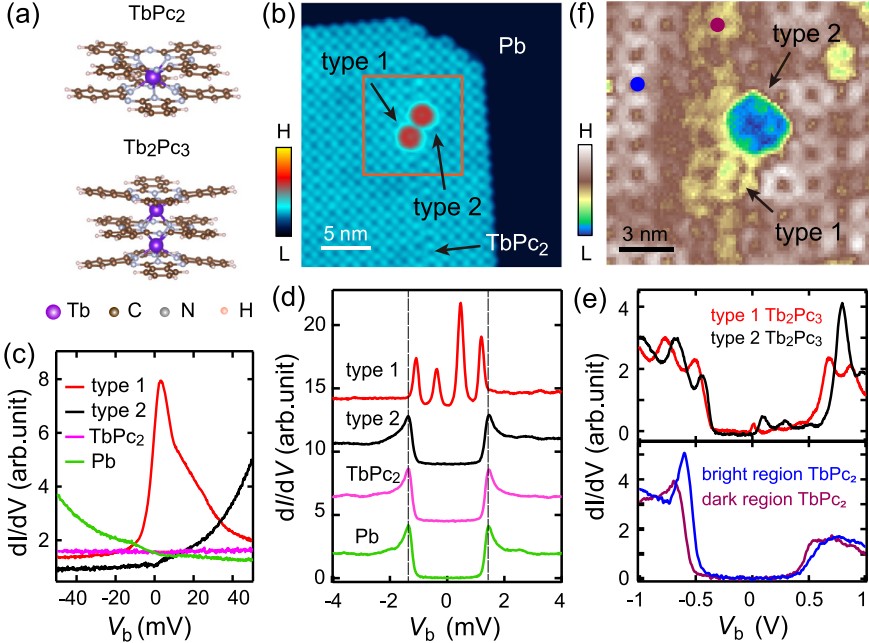

**Fig. 1 | Sample morphology and molecular spin states. a** Molecular structure of $TbPc_2$ and $Tb_2Pc_3$. **b** STM topography ($V = -1.0$ V and $I = 10$ pA) of $TbPc_2$ molecular film with imbedded $Tb_2Pc_3$ on Pb. **c** Tunneling spectra ($V = 50$ mV, $I = 100$pA) of $Tb_2Pc_3$ under 2 T. **d** High-energy resolution spectra ($V = 10$ mV and $I = 100$ pA) of Pb substrate, $TbPc_2$ and the two types of $Tb_2Pc_3$. The dashed lines mark the energy of the superconducting coherence peaks. **e** Tunneling spectra ($V = 1.0$ V and $I = 100$ pA) showing electronic structure of $Tb_2Pc_3$ and $TbPc_2$ at different moiré regions. **f** d$I$/d$V$ mapping ($V = 615$ mV, $I = 100$ pA) of the rectangle area in (**b**).

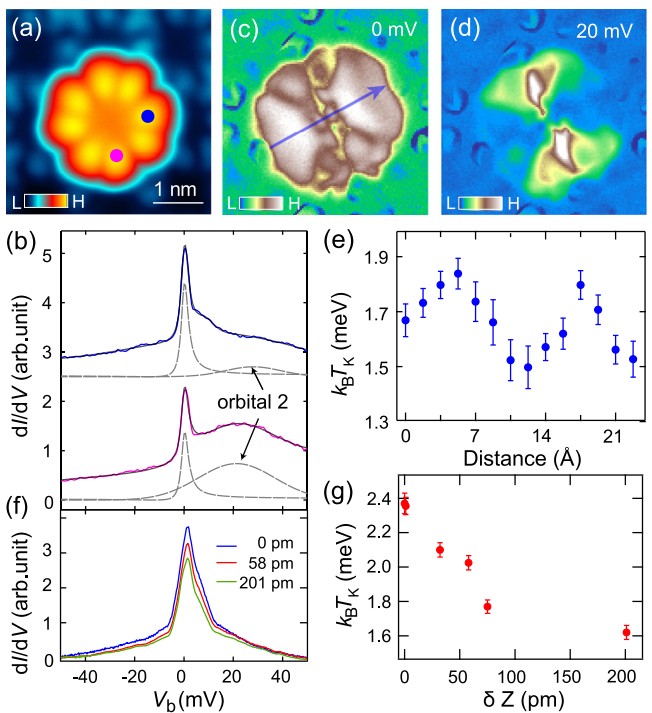

**Fig. 2 | Kondo effect of Tb$_2$Pc$_3$. a** STM image ($V = -1.0$ V and $I = 10$ pA) of an imbedded type-1 Tb$_2$Pc$_3$. **b** Tunneling spectra ($V = 50$ mV and $I = 100$ pA) taken at locations indicated in (**a**) at 2 T, showing Kondo resonances and the upper LUMO. The fitting curves are shown in gray. **c**, **d** d$I$/d$V$ mapping of (**a**) taken at the indicated energies. **e** Variation of the resonance width extracted from the spectra taken along the blue arrow in (**c**). **f** Tunneling spectra taken at the same location of a Tb$_2$Pc$_3$, showing evolution of the Kondo resonance with decreasing tip height δ$Z$, where δ$Z = 0$ pm is set at $V = 5$ mV and $I = 50$ pA. **g** The Kondo peak width extracted from (**a**) as a function of δ$Z$.

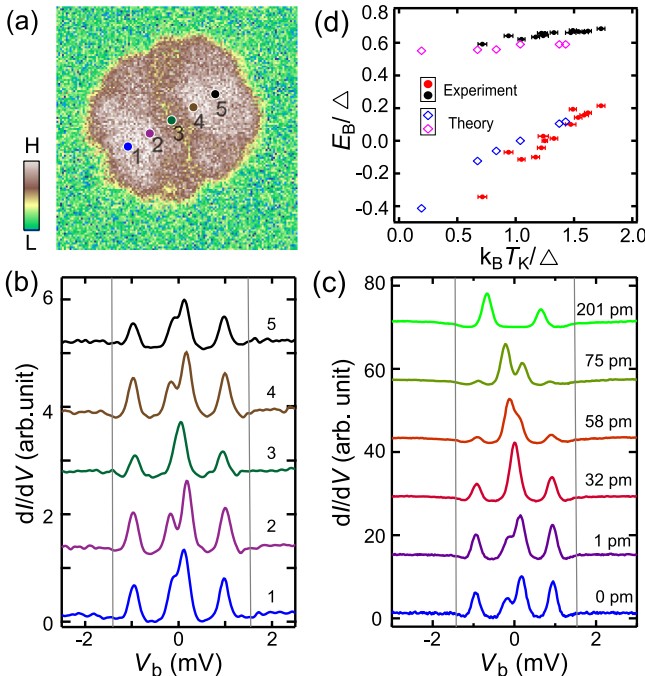

**Fig. 3 | YSR state of Tb$_2$Pc$_3$. a** d$I$/d$V$ mapping of the Tb$_2$Pc$_3$ in Fig. 2a at 0.3 mV, showing the intensity distribution of the YSR state. **)** Selective spectra ($V = 10$ mV and $I = 100$ pA) taken at the indicated positions in (**a**). The gray lines depict the energies of the superconducting coherence peaks of Pb. **c** Variation of YSR states taken at the same location of a Tb$_2$Pc$_3$ with decreasing tip height δ$Z$. The δ$Z = 0$ pm value is set at $V = 10$ mV and $I = 50$ pA. **d** Statistics showing the relation between $E_b$ and $T_K$ relative to the Pb superconducting gap Δ. Black (red) solid dots are experimental outer (inner) YSR state. Pink (blue) hollow dots are theoretically calculated outer (inner) YSR states.

~100 meV (Fig. 1e). This demonstrates the moiré modulation of molecule-substrate interaction[29], resulting in the different charge transfer to Tb$_2$Pc$_3$. The graphene substrate has no effect on the charge transfer process, because its carrier density is negligible compared to that of Pb.

Similar to TbPc$_2$, the STM image of Tb$_2$Pc$_3$ displays an eight-lobed structure corresponding to its top Pc (Fig. 2a). Spectra measured over all the lobes exhibit a Kondo resonance, conforming to the delocalized ligand spin (Fig. 2b). Notably, its LUMO appears at ~20 meV. The LUMO intensity anti-correlates with that of the Kondo resonance (Fig. 2b), as is more explicitly seen from the spectroscopic mapping at 20 and 0 mV, respectively (Fig. 2c, d). Surprisingly, the spatial symmetry of both mappings reduces to two-fold, in apparent conflict with the four-fold symmetry of free Tb$_2$Pc$_3$, suggesting the essential role of substrate coupling. For the type 1 Tb$_2$Pc$_3$, their different embedded positions in the moiré pattern result in the variations in their LUMO energies and the Kondo temperatures, due to the different coupling strength to the substrate. The Kondo peak width, as extracted from the Fano fitting (Supplementary Fig. S5), varies from 1 to 3 meV for different molecules. The Kondo temperature becomes higher when the LUMO is closer to $E_F$, in line with the theoretical expectation[30].

Moreover, within a single Tb$_2$Pc$_3$ the Kondo resonance width exhibits prominent intramolecular variation (Fig. 2e), even though the width is expected to be uniform across the molecule. The variation is most likely a tip effect: the force between the STM tip and the molecule changes the molecule-substrate coupling strength and the magnitude of this effect depends on the lateral position of the tip. This scenario is indeed justified from the decrease of the resonance width at smaller tip height (Fig. 2f, g), which demonstrates that the tip-molecule

interaction lifts the molecule away from the substrate, reducing the molecule-substrate coupling, as observed in related systems[31,32]. Similar effect of the tip force has also been observed on an oxygen-up VOPc molecule, whose interaction with the Pb substrate becomes enhanced with decreasing tip height, inducing quantum phase transition of the associated YSR state[33].

Since Kondo screening competes with the superconducting pairing[21,33,34], the tip-position dependent Kondo coupling provides a way of studying such competing ground states within a single molecule. As shown in Fig. 1d, the YSR states of the type 1 Tb$_2$Pc$_3$ taken at the indicated lobe (Fig. 2a, blue dot) exhibit two prominent YSR pairs located at ±0.48 and ±1.2 meV, respectively. The superconducting coherence peaks on the Kondo molecule are absent due to the transfer of the spectral weight to these subgap states. There is a sizeable asymmetry in the intensity of the YSR states, with a ratio between the hole and electron components of 2.73 (1.43) for the inner (outer) YSR pair. This asymmetry is caused by the non-integer filling of the orbitals which is equivalent to potential scattering.

To unveil the spatial distribution of the YSR states, we performed spectroscopic mapping of the molecule at 0.3 mV, corresponding to the hole component of the inner YSR pair. Interestingly, the YSR state intensity also exhibits a two-fold symmetry (Fig. 3a), similar to that of the Kondo resonance, indicating the common origin of the two features. Figure 3b shows a series of spectra measured along the molecule. The inner and outer YSR pairs are shifted, but in very different ways, both in the amplitude and the direction of the shift. Specifically, the inner (outer) YSR pair shifts nonmonotonically (monotonically) towards $E_F$ with the spectroscopic locations moving from the edge of the molecule to its center. The inner YSR pair has a much larger energy shift of 0.2 meV than that of the outer YSR pair, i.e., 0.06 meV. Similar

behavior exists in other molecules with different Kondo temperatures (Supplementary Fig. S6). More interestingly, the YSR states can be manipulated via changing the tip-sample separation $\delta Z$. As is shown in Fig. 3c, with reducing $\delta Z$, the inner YSR state exhibits even larger intramolecular shift, to the point that the states cross $E_F$ which indicates a quantum phase transition. At the same time, the outer YSR pair stays almost unshifted. This demonstrates that changing $\delta Z$ can effectively tune the strength of the molecule-substrate coupling, conforming to the variation of $T_K$ shown in Fig. 2e, g.

The YSR energy evolves with the Kondo coupling strength, as parametrized by the Kondo temperature $T_K$. The strength of the Kondo coupling $T_K$ relative to the superconducting gap $\Delta$ determines whether the ground state is a Kondo-like singlet or a free-spin-like doublet with a quantum phase transition at $T_K \sim 0.3\Delta$ for spin $S = 1/2$ systems[34,35]. We have collected the statistics on the inner YSR energy and the Kondo temperature scaled by the superconducting gap $\Delta$ on different molecules imbedded inside the TbPc$_2$ films in conjunction with the intramolecular variation. As shown in Fig. 3d, their relation collapses onto the same curve. Namely, the inner YSR state monotonically shifts from negative energy to positive energy with increasing $T_K$ with a quantum phase transition at $T_K \sim 1.26\Delta$. The transition point is larger than that of a conventional spin 1/2 system, calling for future theoretical understanding. On the other hand, the binding energy of the outer YSR state merely slightly increases with increasing $T_K$.

To unravel the origin of the two-pair YSR states, we carried out density functional theory (DFT) plus $U$ calculations[36] for Tb$_2$Pc$_3$ on Pb(111) with the adsorption configuration shown in Fig. 4a. Such configuration conforms to that of an isolated Tb$_2$Pc$_3$ determined experimentally (Supplementary Fig. S7), which displays a Kondo resonance and thus is representative of the type 1 Tb$_2$Pc$_3$. After structure optimization, the total magnetic moment of the molecule is $0.38\mu_B$, demonstrating charge transfer to the LUMO from the substrate, as is also supported from the differential charge distribution (Supplementary Fig. S8). The spatial distribution of the magnetic moment (Fig. 4a) matches that of the Kondo peak. The adsorption induces molecular distortion, lifting the doubly degenerated LUMO and reducing their symmetry from four-fold to two-fold. From the comparison of the DOS projected on the $p_z$ orbitals of C and N atoms (Fig. 4b) with that of the isolated [Tb$_2$Pc$_3$]$^-$ (Supplementary Fig. S9), we assign the two peaks marked with black arrows in Fig. 4b to the degeneracy-lifted LUMO, whose spatial distributions are shown in Fig. 4c. Evidently, orbital 1 of the lower LUMO resembles nicely with that of the spin distribution in Fig. 4a, demonstrating it hosts the ligand spin. Orbital 2 partially overlaps with $E_F$, putting it in the valence fluctuation regime.

The situation in this molecule resembles that in double quantum dots connected to superconductor leads, where two pairs of YSR states have been predicted[37]. Therefore, we model the YSR states with the orbital splitting based on the following Hamiltonian:

$$H = \sum_{k\sigma i}\varepsilon_k c^{\dagger}_{k\sigma i}c_{k\sigma i} + \sum_{ki}\Delta(c^{\dagger}_{k\uparrow i}c^{\dagger}_{-k\downarrow i} + h.c.) + \sum_{i\sigma}\varepsilon_i d^{\dagger}_{i\sigma}d_{i\sigma}$$
$$+ \sum_i U_i n_{i\uparrow}n_{i\downarrow} + \left(\sum_{ik\sigma}V_i d^{\dagger}_{i\sigma}c_{k\sigma i} + h.c.\right) - JS_1 \cdot S_2 \tag{1}$$

Here the first two terms represent the BCS superconducting state of Pb. The third and fourth terms represent the two ($i = 1,2$) ligand orbitals of Tb$_2$Pc$_3$. The fifth term is the hybridization between the molecular orbitals and the Pb substrate, where the hybridization strength is $\Gamma_i^S = \pi V_i^2 \rho_i$. The final term corresponds to the coupling between the spins in the two orbitals. Other notations are standard. The STM tip height determines the substrate-molecule and molecule-tip distances. The hybridization of the molecular orbitals with the tip is much smaller than that with the substrate, $\Gamma_i^T \ll \Gamma_i^S$, thus the YSR energies are determined by $\Gamma_i^S$, while $\Gamma_i^T$ determines the contribution of both orbitals to the total tunneling current, which is calculated via Bardeen's approach from the orbital spectral functions. The model is schematically shown in Fig. 4d. The impurity problem is solved using the NRG method[38,39]. In this model, the ratios $\Gamma_i^S/U_i$ governs the physical properties. We consider the situation in which the 1st orbital is in the local-moment regime with $U_1/\pi\Gamma_1^S > 1$, and the 2nd orbital in the valence-fluctuation regime with $U_2/\pi\Gamma_2^S < 1$. While the spin in a valence-fluctuating orbital is not well defined, it still creates YSR states[40-43], because of its wavefunction component with single electron occupancy. The calculations were performed in the wide-band limit for parameters $U_1/\Delta = 100$, $U_2/\Delta = 10$, $\varepsilon_1/\Delta = -25$, $\varepsilon_2/\Delta = -4$, with fixed large hybridization for $\Gamma_2^S/U_2 = 0.375$ and variable hybridization for $\Gamma_1^S/U_1$ in the range 0.06–0.092. Furthermore, we set $J/U_1 = -0.005$, which corresponds to a weak yet non-negligible Hund's coupling, presumably due to the nearly orthogonal symmetry of the two LUMO states. We assume that both $\Gamma_1^S$ and $\Gamma_1^T$ exponentially depend on $Z$, i.e., $\Gamma_1^S \sim e^{-\kappa_1^S Z}$ and $\Gamma_2^T \sim e^{-\kappa_2^T Z}$, and that $\kappa_2^T/\kappa_1^S = 12$. This empirical model is based on the measured d$I$/d$V$ data, showing variation of inner peak positions, while the outer peak positions are nearly constant. The simulation results shown in Fig. 4e reproduce the measurements from Fig. 3c rather well, supporting our model. This indicates that the valence-fluctuating orbital experiences inter-orbital coupling to the local moment, and the inner (outer) YSR pair originates from spin (orbital), because it closely (slightly) evolves with the changing Kondo temperature. As such, this new type of YSR excitation is dubbed as spin-orbit YSR state. The spin-orbit YSR state should be generic to molecules experiencing orbital degeneracy lifting and charge transfer

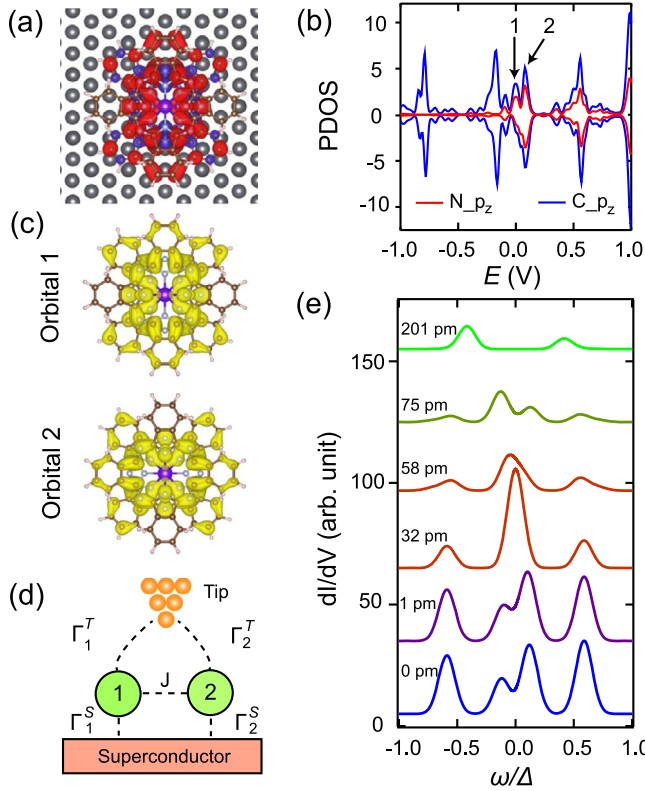

**Fig. 4 | Theoretical calculations. a** DFT calculated spatial distribution of magnetic moment in Tb$_2$Pc$_3$ on Pb(111) (gray balls). The red (blue) color represents spin-up (spin-down) electron density. **b** PDOS of Tb$_2$Pc$_3$ on Pb(111) projected on $p_z$ orbital of C and N atoms. Orbital 1 and orbital 2 are split due to adsorption on Pb(111). **c** Charge distribution of the two orbitals in isolated [Tb$_2$Pc$_3$]$^-$, which correspond to the orbitals marked with arrows in (**b**). **d** Schematics showing the theoretical model. The two orbitals 1 and 2 couple to each other with the exchange coupling $J$, while $\Gamma_i^T$ ($\Gamma_i^S$) represents the hybridization between the orbital $i = 1$, 2 and the tip (superconducting substrate). **e** Calculated variation of YSR states by reducing tip height $\delta Z$.

upon substrate adsorption. The simulation also highlights the dual role of the tip height in this system, affecting either molecule-substrate or molecule-tip coupling for either orbital.

It is noted that our model is formally similar to that of ref. 8, but the parameter regime is entirely different. In the case of ref. 8, the two orbitals strongly couple into a $S = 1$ collective state, resulting in identical particle-hole intensity pattern in their two YSR pairs. In contrast, our case is in the regime of two weakly coupled levels, whose associated two YSR pairs may exhibit different particle-hole asymmetry. Our study indicates that valence fluctuations can suppress the effect of exchange coupling between the spin channels within the same molecular structure, which is a priori expected to be usually strong.

In summary, we have investigated the two-pair YSR states in $Tb_2Pc_3$ on Pb (111). The LUMO of $Tb_2Pc_3$ gets singly occupied upon adsorption on Pb, and experiences orbital degeneracy lifting. The ligand spin induces Kondo screening and YSR subgap states in the Pb host. The orbital splitting of the LUMO and its charge fluctuation results in two pairs of YSR states, corresponding to a weakly-coupled two-impurity YSR system in a single molecule. Similar YSR state has also been observed in $Y_2Pc_3$, confirming the negligible influence of the f-moment on the ligand spin state, in agreement with DFT calculations on $Tb_2Pc_3$ with full f-electrons plus $U$ (Supplementary Note 2). Our work identifies a new category of spin-orbital YSR state, which permits a study of spin physics in a multi-level molecule that would otherwise be difficult to spectrally resolve and may aid the understanding of more complex many-body excitations. In view of the delocalized nature of the ligand spin, this also paves the way for tailoring magnetically coupled molecular structures for realizing topological superconductivity.

## Methods
### Experiments
The experiments were carried out with a custom-designed Unisoku STM (1300) at 0.4 K. The detailed procedures for preparing the graphene covered SiC(0001) substrate have been reported in ref. 44. The Pb mesa is grown on the graphene substrate at room temperature. The thickness of the grown Pb islands is typically about 5 nm, whose superconducting gap size at 0.4 K is measured as about 1.35 meV (Supplementary Fig. S10), which is close to the bulk value and also conforms to the previous study[45]. $TbPc_2$ molecules are evaporated onto the Pb surface at room temperature. An electro-etched W wire was used as the STM probe, which had been cleaned and characterized on a Cu(111) crystal prior to the measurement. The d$I$/d$V$ spectra were acquired using conventional lock-in technique with a bias modulation of 1 mV and 50 μV at 983 Hz for the detection of Kondo peaks and YSR states, respectively.

### Theory
The DFT calculations are carried out by Vienna ab-initio simulation (VASP) package[46,47] with the projected augmented wave (PAW) method[48]. Exchange-correlation functional with non-local correction via van-der Waals interaction developed by Hamada[49] were used. VASPKIT package[50] was used to analyze the DOS calculation data. The molecular structures, charge densities, and spin distributions are visualized by VESTA[51]. The Pb substrate is modeled by 4-layer thickness slab with the supercell size of $(8 \times 8)$. To stabilize the ligand spin, we employ $U = 4.5$ eV and $J = 1.0$ eV on $p$-orbitals of N and C atoms. We used pseudopotential of Tb(III) ion in VASP with a fixed f-electron state, which is a standard method for assessing the ligand electronic state of the phthalocyanine complex[52]. This treatment is rationalized by additional calculations on $Tb_2Pc_3$ using full f-electrons plus $U$, which deliver similar electronic states of molecular levels as the fixed f-electron pseudopotential (Supplementary Note 2). This conforms to the strong localization of Tb f-electrons in the phthalocyanine complex, which makes the interaction between f- and π-electrons weak[53,54].

In the calculations of $Tb_2Pc_3$ on Pb(111), the initial guess of the magnetic moment for all the carbon and nitrogen atom was set to 0.1 $\mu_B$ (in total 12.0 $\mu_B$). Therefore, the obtained solution was started from a state without time-reversal symmetry. We carried out electronic and structure relaxation without any constraint on the magnetic state. Thus, our calculation reflects the ground state. Due to the structural symmetry breaking of $Tb_2Pc_3$ on Pb(111), one of the LUMO orbitals possesses the unpaired electron, and the other orbital has much smaller occupation. The AFM coupling between the two orbitals in $Tb_2Pc_3$ on Pb(111) could be seen from the spin density for the majority and minority spin components in Supplementary Fig. S11. The minority spin component is orthogonal to, and smaller than, the majority spin component. Considering that the two LUMO orbitals are orthogonal, this suggests that the first (second) LUMO orbital is occupied by majority (minority) spin, and consequently there is AFM coupling between the two orbitals. We note that the Pb(111) substrate plays an important role to form the AFM coupling, and the magnetic coupling in an isolated $Tb_2Pc_3$ may be different.

The NRG calculations were performed with the NRG Ljubljana package (https://github.com/rokzitko/nrgljubljana). The discretization parameter was $\Lambda = 4$, we kept up to 5000 multiplets (the sole conserved quantum number was the total spin S), or cutoff 9 in units of the current NRG scale. The subgap part of the spectral functions was broadened using Gaussian kernels with width $\Delta/10$, thereby mimicking the experimental broadening due to finite temperature and electromagnetic noise. The tunneling current is simulated via Bardeen's tunneling approach:

$$I = 4\pi e/\hbar \sum_{i=1}^{2} \int_0^{eV} \rho_{S,i}(\mu - eV + \delta\epsilon)\rho_T(\mu + \delta)|M_i|^2 d\delta.$$

Here the sum runs over the two molecular orbitals that contribute most current in the subgap region; we assumed that there is no interference between the two contributions. $V$ is the bias voltage, $\rho_{S,i}$ is the spectral function of the orbital $i$ (computed using the NRG solver), $\rho_T$ is the tip spectral function (assumed constant in energy), and $M_i$ is the tunneling matrix element between the tip and the molecular orbital $i$. These matrix elements are proportional to the tip hybridization function $\Gamma_i^T$ (see Fig. 4d for a schematic representations of the couplings) and we assume them to be constant in energy. We furthermore assume that the hybridizations $\Gamma_1^S$ and $\Gamma_2^T$ depend exponentially on the tip position, as in $\Gamma_1^S = \Gamma_1^S(0) \exp[-\kappa_1^S(Z - Z_0)]$ and $\Gamma_2^T = \Gamma_2^T(0) \exp[-\kappa_2^T(Z - Z_0)]$, while other hybridizations do not depend on $Z$. We find that good agreement with the experimental measurements is obtained for $\kappa_2^T/\kappa_1^S = 12$.

### Molecule synthesis
Synthesis of $TbPc_2$ is accomplished by direct cyclization of the ring pre-cursor (1,2-dicyanobenzene) at high temperatures in the presence of a metal salt (e.g., Tb(acac)3), 1,8-diazabicyclo (5.4.0) undec-7-ene (DBU) and high boiling solvents (e.g., penthanol, hexanol)[55].

## Data availability
All data needed to evaluate the conclusions in the paper are present in the paper and the Supplementary Information. Additional data related to this paper may be requested from the authors.

## Code availability
The code that supports the findings of this study is available from the corresponding author upon reasonable request.

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

## Acknowledgements

We thank Prof. Michal Otepka, Prof. Pavel Jelinek, and Prof. Guoying Gao for helpful discussions. This work was funded by the National Key Research and Development Program of China (Grant Nos. 2017YFA0403501, 2018YFA0307000), the National Natural Science Foundation of China (Grant Nos. U20A6002, 11874161, and 11774105). R.Ž. acknowledges the support of the Slovenian Research Agency (ARRS) under P1-0044 and P1-0416. DFT calculations were performed using a computer facility at the Research Center for Computational Science (Okazaki, Japan).

## Author contributions

H.N.X., Z.Y.L., and X.L. did the experiments with the help of M.C., Z.H.L., and W.H.Z.; E.M. did the DFT calculations; R.Z. did the NRG modeling; S.K. and M.R. synthesized the molecule; Y.S.F., H.N.X., E.M., R.Z., and M.R. analyzed the data and wrote the manuscript with comments from all authors. Y.S.F. supervised the project.

## Competing interests

The authors declare no competing interests.

## Additional information

**Correspondence and requests** for materials should be addressed to Ying-Shuang Fu.

