## [Peer Review File · Nature Communications]

Reviewers' Comments:

Reviewer #1:

Remarks to the Author:

This submitted interesting paper reports on the observation of the two-pair YSR states in Tb₂Pc₃ on Pb (111). The claimed reason of it is related to the split of the degenerate LUMO in Tb₂Pc₃ upon adsorption on Pb and with a consequent single population of the lowest component due to charge transfer mechanism. In such a framework, this ligand spin induces Kondo screening and YSR subgap states in the Pb host. Therefore, the two observed pairs of YSR states are originated by the orbital splitting of the LUMO and its charge fluctuation results, corresponding to a weakly-coupled two-impurity YSR system in a single molecule.

The presented paper is surely interesting but I have several concerns and comments related to it:

1) In general, a more detailed description of what the authors have done both at the experimental and theoretical aspects should be presented. The further information would help a lot the readability and the understanding of what has been done dispelling any possible doubt.

2) In this regard, more details on the formation of the Tb₂Pc₃ should be given. In particular, when they say " Tb₂Pc₃ ..., which are formed via the cracking of TbPc₂ into TbPc and a subsequent chemical reaction with another TbPc₂": how does the cracking take place?

3) they present a portion of an island that they claim is being formed by cracked Pc and TbPc. A comment on the expected charge of these two species should be added but this it puzzles me is the stability of the TbPc. The way that such a fragment with a large lanthanide ion and with half of its coordination sphere unsaturated can interact with the surface can be only two: one where the lanthanide somehow saturates its coordination sphere with the lead density; the other where the Pc ligand is adsorbed on the surface with the Tb ion exposing its unsaturated coordination sphere.

In Figure S1, the dimensions are not the same as the ones reported in Fig S1 a) and b) and, therefore, it does not help that much. For these reasons, I cannot exclude that the bright spots (supposed to be a TbPc species by height considerations) can be the exposed Tb(III) ion on top of a Pc@Pb or the Pc ligand on top of the Tb(III)@Pb.

This point is a crucial one because could lead to the question of whether the observed peaks in the STM height profiles can be derived by a PcTbPcTbPc@Pb scenario or to the PcTbPcPcTb@Pb one.

Related to this point can help the isolated molecule extracted by the island (Figures S1-2). In this case, it is likely that it can be a PcTbPcTbPc@Pb case but the authors do not specify if they perform the Kondo measurements on stand-alone molecules or extracted by the islands. I say this because authors should provide relevant statistics to prove that the extracted molecules, supposed to be Tb₂Pc₃, show YSR states. In this form, the paper does not provide it. The authors should provide convincing data that observed brighter spots observed are the same species.

4) The authors should also give more details about DFT calculations that are missing in the actual form of the paper. They say that found a 0.38 μB on the PcTbPcTbPc@Pb geometry: Do they perform the calculations on an antiferromagnetic state? If yes, they should provide the solution of the Broken State solution. Ferromagnetic solution should be also computed to verify if the AF is the ground state. Moreover, the PcTbPcPcTb@Pb scenario should be also tested.

In such a framework, STM simulated images are mandatory to disentangle the two scenarios.

5) the proposed scheme in Figure 4d should be made clearer.

6) There are missing symbols on page 9.

On the basis of the above comments and concerns, I recommend major revisions so that the paper can be suited for publication in Nat. Comm.

Reviewer #2:

Remarks to the Author:

Molecular YSR states have attracting increasing attention in the last few years. Double and triple decker phthalocyanine molecules have been interesting Kondo systems driven by a single electron on the phthalocyanine ligand. The present paper now combines interest in specific molecular system with YSR physics on a superconductor. The authors observe YSR states in some triple decker molecules.

Despite of the potential interest in the system, I do not find substantial new insights from this study. The authors claim to have discovered a new type of YSR states, which they call spin-orbital YSR states. However, the fact that the orbital is in the valence-fluctuation regime does not impose significant changes to the exchange coupling of the spin. As known from quantum-dot systems, the partial occupation of a state leads to exchange coupling. Coupling to an additional orbital has also been observed previously. In fact, the associated Hamiltonian is the same as in Rubio-Verdu et al. (Ref. 8), where authors already describe coupling of spins in different molecular orbitals. Thus, the paper does not present important advances as expected for publications in Nature Communications.

Reviewer #3:

Remarks to the Author:

The manuscript reports on a novel mechanism in molecular systems behind the onset of Yu-Shiba-Rusinov (YSR) states. Tb₂Pc₃ molecules deposited, along with TbPc₂, on a Pb superconducting substrate are investigated by means of conductance measurements. Although Tb₂Pc₃ does not possess spare electrons on its ligands, a charge transfer process activate the ligand spin. The authors show the presence of a Kondo peak and, in presence of the Pb superconducting phase, YSR states. These features originate from a spin localized on the organic ligands of some of the Tb₂Pc₃ molecules. Additionally, this charge transfer is modulated by a moiré pattern that explains the different behavior of the Tb₂Pc₃ molecules on the surface (type 1 and type 2). Due to the presence of the surface the four-fold symmetry of the molecule is reduced to a two-fold symmetry lifting the LUMO degeneracy. Interestingly, the presence of a pair of two YSR states of the spin activated Tb₂Pc₃ originates from the degeneracy-lifted LUMO and are related to the interplay between a local moment and a valence fluctuating orbital.

Their claims are novel and of great interest. The data presented in support of their conclusions are of high quality. However, I have some points that need to be clarified before I can recommend it for publication:

One of the key ingredient for the observation of the Kondo and YSR states feature in dI/dV data is the presence of a charge transfer between the surface and the Tb₂Pc₃ molecules. While the presence of such a mechanism is well supported, additional clarifications would be beneficial.

a) Do authors observe an effect of the molecular orientation respect to the Pb surface?

b) The authors use as substrate a Pb film deposited on graphene covered SiC(0001) surface. Is the presence of the underneath graphene layer relevant for the charge transfer process? Does the charge transfer also depend on the film thickness?

The superconducting properties of the deposited Pb depend on the thickness of the layer and on the dimension of possible islands [Fabian Paschke, *adv quantum technol.* 2020, 3, 2000082]. Are the measurements acquired on specific islands or the Pb forms a complete film? I think the authors should clarify these points.

In the case reported by the manuscript, the charge fluctuation phenomenon at the origin of the two pair of YSR states of type 2 Tb₂Pc₃ involves the lift of the orbital degeneracy of the LUMO but how general is this charge fluctuation phenomenon? What are the requirements to observe such a phenomenon on a single molecule scale? I believe that a comment on this aspect would help to clarify the relevance of this work.

Minor points:

- In Fig.1d the grey lines related to the coherence peaks are misplaced.
- Page 7 the authors refer to Fig.2 g and h: "[...] conforming to the variation of TK shown in Figs. 2(g,h)." there is no panel h in Fig.2.
- Labelling of data in Fig.S5 is inconsistent between panel a and c.

Reply to review

Reviewer 1:

Comments:

This submitted interesting paper reports on the observation of the two-pair YSR states in Tb_2Pc_3 on Pb (111). The claimed reason of it is related to the split of the degenerate LUMO in Tb_2Pc_3 upon adsorption on Pb and with a consequent single population of the lowest component due to charge transfer mechanism. In such a framework, this ligand spin induces Kondo screening and YSR subgap states in the Pb host. Therefore, the two observed pairs of YSR states are originated by the orbital splitting of the LUMO and its charge fluctuation results, corresponding to a weakly-coupled two-impurity YSR system in a single molecule.

Reply: We thank the referee for commenting on the interesting study of our paper. The referee's comments certainly help to improve our manuscript.

(1) In general, a more detailed description of what the authors have done both at the experimental and theoretical aspects should be presented. The further information would help a lot the readability and the understanding of what has been done dispelling any possible doubt.

Reply: We thank the referee for this important suggestion. We have added more detailed descriptions in the revised version to clarify the unclear issues, where major changes to our manuscript have been marked in red and summarized in the list of changes.

(2) In this regard, more details on the formation of the Tb_2Pc_3 should be given. In particular, when they say “ Tb_2Pc_3 ..., which are formed via the cracking of $TbPc_2$ into $TbPc$ and a subsequent chemical reaction with another $TbPc_2$ ”: how does the cracking take place?

Reply: This is an important point. The spontaneous formation of Tb_2Pc_3 from $TbPc_2$ precursors via sublimation in ultrahigh vacuum (UHV) has been studied by some of us in *Nanoscale* 10, 15553-15563 (2018) with high-resolution mass spectrometry, AFM and STM. According to that study, $TbPc_2$ can be decomposed into $TbPc$ and Pc , which subsequently form Tb_2Pc_3 at a high temperature of 650-700 K **in the crucible**. Such reaction follows the pathway:

Where the growth temperature ΔT and the time Δt affect the total amount of molecules participating in the reaction. High-resolution mass spectrometry for residual material remaining in the crucible after the UHV experiments, confirms the presence of $TbPc_2$, Tb_2Pc_3 , and the half-decker species. Furthermore, the ratio between $TbPc$ and Pc estimated from Fig. S1(d) is 1:1.8, which is close to the value of 1:2 expected from the above reaction pathway.

For clarity, we have added two sentences starting with “The Tb_2Pc_3 molecules are formed...” to Page 4 of main text, and three sentences starting with “The Tb_2Pc_3 is...” to supplementary note 1.

(3) they present a portion of an island that they claim is being formed by cracked Pc and $TbPc$. A comment on the expected charge of these two species should be added but the this it puzzles me is the stability of the $TbPc$. The way that such a fragment with a large lanthanide ion and with half of its coordination sphere unsaturated can interact with the surface can be only two: one where the lanthanide somehow saturates its coordination sphere with the lead density; the other where the Pc ligand is adsorbed on the surface with the Tb ion exposing its unsaturated coordination sphere.

In Figure S1, the dimensions are not the same as the ones reported in Fig S1 a) and b) and, therefore, it does not help that much. For these reasons, I cannot exclude that the bright spots (supposed to be a $TbPc$ species by height considerations) can be the exposed $Tb(III)$ ion on top of a $Pc@Pb$ or the Pc ligand on top of the $Tb(III)@Pb$.

This point is a crucial one because could lead to the question of whether the observed

peaks in the STM height profiles can be derived by a PcTbPcTbPc@Pb scenario or to the PcTbPcPcTb@Pb one. Related to this point can help the isolated molecule extracted by the island (Figures S1-2). In this case, it is likely that it can be a PcTbPcTbPc@Pb case but the authors do not specify if they perform the Kondo measurements on stand-alone molecules or extracted by the islands. I say this because authors should provide relevant statistics to prove that the extracted molecules, supposed to be Tb₂Pc₃, show YSR states. In this form, the paper does not provide it. The authors should provide convincing data that observed brighter spots observed are the same species.

Reply: Thanks for raising this crucial point. The referee's central concern is whether the bright molecules showing the Kondo resonance and the YSR states are Tb₂Pc₃ or TbPc₂ on TbPc. Those two configurations, according to the referee's designation, correspond to PcTbPcTbPc@Pb and PcTbPcPcTb@Pb, respectively.

We provide the following evidence to justify our Kondo molecules are Tb₂Pc₃. First, we have performed STM tip manipulation to the bright molecule, as shown in Figs. S1a,b. The manipulated bright molecule appears as a robust unit, without decomposing. This observation is contradictory to the proposed PcTbPcPcTb@Pb stacking, where PcTbPc is only pi-pi interacting with lower PcTb@Pb and cannot be laterally manipulated together with the upper PcTbPc part. This rigorously proves the bright molecule is Tb₂Pc₃. Second, Tb₂Pc₃ molecules could form inside the crucible through chemical reaction with TbPc₂ precursors, as depicted in our reply to Question 2. Thus, Tb₂Pc₃ molecules could be directly sublimed onto the Pb surface, rendering the available configuration of PcTbPcTbPc@Pb. Third, TbPc molecules appear with bright centers in their STM images of Fig. S1d, suggesting their adsorption configuration as TbPc@Pb. Otherwise, STM topography of the PcTb@Pb configuration wouldn't have the bright molecular center, similar to that of TbPc₂. The TbPc@Pb configuration makes the scenario of PcTbPcPcTb@Pb less likely.

The adsorption configuration of TbPc on the Pb surface can be further clarified with

DFT calculations. We evaluated the cross-sections of the charge density as the reference for STM topographic image. As shown in Fig. R1, the calculation result of TbPc@Pb (Figs. R1a-c) matches that of the experimental image nicely and is distinct from that of PcTb@Pb (Figs. R1d-f). We also confirmed the charge state in the TbPc/Pb(111) configuration. In this study, we used a Tb potential in which the 4f-electrons are treated as core electrons and their energy levels are fixed to those in $(4f)^8$ state so that the electronic configuration in Tb(III) state is reproduced. Note that this potential is officially provided by VASP and widely used in the calculation of Tb complex. In this setting, outermost 6s state plays important role to determine the charge state of the Tb ion. We confirmed that the occupancy of the 6s state of the Tb ion is 0.34 in TbPc/Pb(111), which is close to that in TbPc₂ (0.27). This supports that the Tb ion is in a stable Tb(III) state $((4f)^8 (6s)^0)$.

Fig. R1 DFT calculation results of TbPc/Pb(111) and PcTb/Pb(111). (a,d) The cross-sections of the charge density of the TbPc/Pb(111) (a) and the PcTb/Pb(111) (d), which correspond to their simulated STM topographic images. The cross-sectional charge densities are obtained on the xy -plane at 4.4 (5.2) Å higher than the Pb(111) surface for the image in (a) [d]. In TbPc/Pb(111) case, the central Tb ion becomes most visible because of the protrusion structure. Meanwhile, in PcTb/Pb(111), the shuttlecock structure makes the outer benzene rings most visible. (b,c) Top (b) and side (c) view of the adsorption configuration of TbPc/Pb(111). (e,f) Top (e) and side

(f) view of the adsorption configuration of PcTb/Pb(111).

For the isolated Tb_2Pc_3 molecules, they also induce Kondo resonances and associated YSR states. Such data is shown in the revised Fig. S7 (the original Fig. S6). About ten isolated Tb_2Pc_3 molecules are extracted out of the film. They all expectedly exhibit Kondo resonances after becoming isolated, irrespective of originally being type-1 or type-2 Tb_2Pc_3 molecules in the film.

The observed bright molecules should be of the same species of Tb_2Pc_3 . This is because their dI/dV spectra show similar features of molecular orbitals, which are slightly shifted by the Moiré pattern, as shown in Fig. 1e. Moreover, the Tb_2Pc_3 molecules, irrespective of being type-1 or type-2 molecules in the film, all exhibit Kondo resonances after becoming isolated.

For clarity, we have made the following changes to our manuscript.

- (1) Supplementary note 1 is revised to include the above discussions.
- (2) Fig. S1d is adjusted to make its scale bar conform to that of Figs. S1a,b.
- (3) Fig. S7 is revised with Kondo resonance and YSR state of isolated Tb_2Pc_3 added.
- (4) Fig. R1 is added to supplementary materials as Fig. S2.

4) The authors should also give more details about DFT calculations that are missing in the actual form of the paper. They say that found a $0.38\mu_B$ on the PcTbPcTbPc@Pb geometry: Do they perform the calculations on an antiferromagnetic state? If yes, they should provide the solution of the Broken State solution. Ferromagnetic solution should be also computed to verify if the AF is the ground state. Moreover, the PcTbPcPcTb@Pb scenario should be also tested. In such a framework, STM simulated images are mandatory to disentangle the two scenarios.

Reply: Thanks for raising this issue. In the DFT calculations for triple-decker Tb_2Pc_3 on Pb(111), the initial guess of the magnetic moment for all the carbon and nitrogen atom was set to $0.1 \mu_B$ (in total $12.0 \mu_B$). Therefore, the obtained solution was started from a state without time-reversal symmetry. We carried out electronic and structure relaxation without any constraint on the magnetic state. Thus, we believe our

calculation reflected the ground state. Due to the structural symmetry breaking of Tb_2Pc_3 on $\text{Pb}(111)$, one of the LUMO orbitals possesses the unpaired electron, and the other orbital has much smaller occupation. The AFM coupling between the two orbitals in Tb_2Pc_3 on $\text{Pb}(111)$ could be seen from the spin density for the majority and minority spin components in Fig. R2. The minority spin component is orthogonal to, and smaller than, the majority spin component. Considering that the two LUMO orbitals are orthogonal, this suggests that the first (second) LUMO orbital is occupied by majority (minority) spin, and consequently the AFM coupling between the two orbitals. We note that the $\text{Pb}(111)$ substrate plays an important role to form the AFM coupling, and the magnetic coupling in an isolated Tb_2Pc_3 may be different.

Fig. R2 **Spin density of $\text{Tb}_2\text{Pc}_3/\text{Pb}(111)$** . Spin density for the majority (a) and minority (b) spin components of Tb_2Pc_3 . The isosurface level is $9.4 \times 10^{-5} \mu_B/\text{Bohr}^3$ for (a) and $9.4 \times 10^{-5} \mu_B/\text{Bohr}^3$ for (b).

As discussed in replies to the comments (2) and (3), the scenario of PcTbPcPcTb@Pb can be excluded. Therefore, we did not carry out the calculations of its magnetic state and STM image simulations.

For clarity, we have added several sentences starting with “In the calculation of ...” to Page 12. And, Fig. R2 is added as Fig. S11 to supplementary materials.

5) the proposed scheme in Figure 4d should be made clearer.

Reply: Thanks for the nice suggestion. We have added captions to Fig. 4d to make it clearer.

6) There are missing symbols on page 9.

Reply: We apologize for the mistake. We have changed the formula of $\Gamma \propto e^{-\kappa Z}$ to $\Gamma_1^S \propto e^{-\kappa_1^S Z}$ and $\Gamma_2^T \propto e^{-\kappa_2^T Z}$ on Page 10 (original Page 9). For other possible missing symbols, we have already specified “Other notions are standard.” in Page 9.

Referee 2:

Comment:

Molecular YSR states have attracting increasing attention in the last few years. Double and triple decker phthalocyanine molecules have been interesting Kondo systems driven by a single electron on the phthalocyanine ligand. The present paper now combines interest in specific molecular system with YSR physics on a superconductor. The authors observe YSR states in some triple decker molecules.

Despite of the potential interest in the system, I do not find substantial new insights from this study. The authors claim to have discovered a new type of YSR states, which they call spin-orbital YSR states. However, the fact that the orbital is in the valence-fluctuation regime does not impose significant changes to the exchange coupling of the spin. As known from quantum-dot systems, the partial occupation of a state leads to exchange coupling. Coupling to an additional orbital has also been observed previously. In fact, the associated Hamiltonian is the same as in Rubio-Verdu et al. (Ref. 8), where authors already describe coupling of spins in different molecular orbitals. Thus, the paper does not present important advances as expected for publications in Nature Communications.

Reply: We thank the referee for commenting on the interesting system of our research. We would like to clarify that the valence fluctuation strongly weakens the overall effect of exchange coupling of spin, and our work unveils a new parameter regime of YSR physics that is distinct to that of Ref. 8. Our reasons are depicted in detail below.

The situation of two weakly coupled levels, one in local moment and another in valence fluctuation regime, has not been studied in the context of YSR states **experimentally**. The new element of our study is that the superconducting gap opens an energy window that permits a study of spin physics in a multi-level molecule that would otherwise be difficult to spectrally resolve.

The model in Ref. 8 formally looks similar, but the parameter regime is entirely different. In the case of Ref. 8, the two orbitals couple into a $S = 1$ collective state, which does not happen in our case. Quite the contrary, we are in the regime of two weakly coupled levels. Because of the distinct parameter regime, the behavior of the YSR states is distinctly different. In Ref. 8, the two pairs of YSR states have identical particle-hole intensity pattern. However, in our case, the two pairs of YSR states may exhibit different particle-hole asymmetry.

Our regime is interesting in itself: one would a priori expect that two spins within the same molecular structure would generally experience strong exchange coupling. It is valence fluctuations that suppress the spin moment and concomitantly the overall effect of exchange coupling, since the latter is a product of the exchange coupling and the coupled spin moments. Our modelling unveils that the Hund's coupling between the two spin levels is of weak yet nonnegligible AFM-type with $J/U_1 = -0.005$. As is experimentally observed in Fig. 3c and theoretically reproduced in Fig. 4e, the two YSR pairs evolve nearly independently with the changing molecule-substrate interaction strength, suggesting the coupling between the two spin levels should be small. Yet, once the inner YSR pair crosses Fermi level E_F , the intensity of the outer YSR pair becomes significantly altered, demonstrating the finite coupling of the two spin levels. The exchange coupling J is constrained in value. If it were large, there would be a significant splitting of YSR peaks (into singlet and triplet components), and also lead to a significant shift of outer YSR peak position when the inner ones cross E_F , due to a change in the ground state of the system across such transition; which is not observed. Based on these considerations, J is estimated to be of order $-0.001U_1$, and certainly smaller than $J = -0.01U_1$ which already leads to visible

singlet-triplet splitting. For the results presented in the manuscript, we take $J = -0.005 U_1$.

To clarify the novelty of our finding and its distinction to the previous study, we have added a few sentences starting with “It is noted that our model...” to Page 10 and a phrase starting with “permits a study of ...” to Page 11 of main text.

Referee 3:

Comment:

The manuscript reports on a novel mechanism in molecular systems behind the onset of Yu-Shiba-Rusinov (YSR) states. Tb₂Pc₃ molecules deposited, along with TbPc₂, on a Pb superconducting substrate are investigated by means of conductance measurements. Although Tb₂Pc₃ does not possess spare electrons on its ligands, a charge transfer process activates the ligand spin. The authors show the presence of a Kondo peak and, in presence of the Pb superconducting phase, YSR states. These features originate from a spin localized on the organic ligands of some of the Tb₂Pc₃ molecules. Additionally, this charge transfer is modulated by a moiré pattern that explains the different behavior of the Tb₂Pc₃ molecules on the surface (type 1 and type 2). Due to the presence of the surface the four-fold symmetry of the molecule is reduced to a two-fold symmetry lifting the LUMO degeneracy. Interestingly, the presence of a pair of two YSR states of the spin activated Tb₂Pc₃ originates from the degeneracy-lifted LUMO and are related to the interplay between a local moment and a valence fluctuating orbital. Their claims are novel and of great interest. The data presented in support of their conclusions are of high quality. Their claims are novel and of great interest. The data presented in support of their conclusions are of high quality. However, I have some points that need to be clarified before I can recommend it for publication:

Reply: We thank the referee for carefully reviewing our manuscript and his/her appreciation to our work. The referee’s comments certainly help us to improve our manuscript.

a) Do authors observe an effect of the molecular orientation respect to the Pb surface?

Reply: Thanks for raising this issue. We have checked the orientations of many molecules and found they all adopt the same orientation as the isolated molecule shown in Fig. S7. Such an orientation is also consistent with our DFT calculated orientation shown in Fig. 4a, which is the most stable adsorption orientation.

For clarity, we have added two sentences starting with “This determines the ...” to the caption of Fig. S7.

b) The authors use as substrate a Pb film deposited on graphene covered SiC(0001) surface. Is the presence of the underneath graphene layer relevant for the charge transfer process? Does the charge transfer also depend on the film thickness? The superconducting properties of the deposited Pb depend on the thickness of the layer and on the dimension of possible islands [Fabian Paschke, *adv quantum technol.* 2020, 3, 2000082]. Are the measurements acquired on specific islands or the Pb forms a complete film? I think the authors should clarify these points.

Reply: Thanks for raising these important issues. The graphene substrate has no effect on the charge transfer process. This is because the graphene layer has a very low carrier density, which is negligible compared to the electron density of the Pb films. Recently, we have also grown Pb on SrTiO₃ substrate. The Tb₂Pc₃ molecules on Pb/SrTiO₃ show the same YSR states as those on Pb/graphene. This also demonstrates the graphene substrate has no effect.

Thank the referee for bring us the attention of the reference of *Adv. Quantum Technol.* 3, 2000082 (2020). According to that paper, the superconducting gap size depends on the thickness of the Pb island, but is independent of the island size. The Pb in our study also forms compact islands, whose thicknesses are typically about 5 nm. Fig. R3 shows the superconducting gap of a typical Pb island with a thickness of 5 nm and a lateral size of 70 nm. BCS-fitting to the superconducting gap determines a gap size of

1.35 meV, which is very close to the bulk value 1.36 meV and also conforms to that reported in the mentioned paper.

We focused on the charge transfer process mediated by the Moiré pattern, instead of the thickness dependence of the Pb films. However, we agree with the referee that the Pb films thickness could be another interesting parameter to tune the charge transfer of the molecule. Because of the quantum well states in the Pb films that are formed via electron quantization perpendicular to the film surface, the work function of Pb should be modulated by the film thickness [Ref. Qi, Y. et al. Appl. Phys. Lett. 90, 013109 (2007)], which would expect to impact the charge transfer to Tb_2Pc_3 . Such an experiment is beyond the scope of the current research, but is indeed under our plan of future investigations.

Fig. R3 (a) BCS fitting of the superconducting gap of a Pb island measured at 0.4 K. (b) Topographic image of a typical Pb island with TbPc_2 and Tb_2Pc_3 molecules on top. The spectrum in (a) was obtained on the Pb island of (b).

For clarity, we have added a sentence “The graphene substrate has no effect on the charge transfer process, because its carrier density is negligible compared to that of Pb.” to Page 5 of main text, and a sentence “The thickness of the grown Pb islands is typically about 5 nm, whose superconducting gap size at 0.4 K is measured as about 1.35 meV (Fig. S10), which is close to the bulk value and also conforms to the

previous study [44].” to methods of Page 11. We have also added Fig. R3 to supplementary materials as Fig. S10, and added a new Ref. 44.

In the case reported by the manuscript, the charge fluctuation phenomenon at the origin of the two pair of YSR states of type 2 Tb₂Pc₃ involves the lift of the orbital degeneracy of the LUMO but how general is this charge fluctuation phenomenon? What are the requirements to observe such a phenomenon on a single molecule scale? I believe that a comment on this aspect would help to clarify the relevance of this work.

Reply: Thanks for raising this important issue. The charge fluctuating orbital should be a general phenomenon. There are many molecules that possess doubly degenerated LUMO orbitals, whose degeneracy can be broken upon adsorption on the substrate due to mismatched symmetry. Usually, there is about one electron transferred onto the molecule from the substrate. This could likely create a molecular orbital in the spin-1/2 regime. If the higher split LUMO orbital is close to the Fermi level, a charge fluctuating orbital forms, which would interact with the spin-1/2 orbital. Thus, our case of Tb₂Pc₃ is certainly not the only example, but should represent a general phenomenon in other similar molecules as discussed above.

For clarity, we have added two sentences starting with “As such, this new type of YSR excitations is dubbed as spin-orbital YSR state. The spin-orbital YSR state should be generic to molecules experiencing orbital degeneracy lifting and charge transfer upon substrate adsorption.” to Page 10 of main text.

In Fig.1d the grey lines related to the coherence peaks are misplaced.

Reply: We apologize for the mistake. We have corrected the lines in the revised version.

Page 7 the authors refer to Fig.2 g and h: “[...] conforming to the variation of T_K shown in Figs. 2(g,h).” there is no panel h in Fig.2.

Reply: We apologize for the mistake. We have corrected the panel in the revised version.

Labelling of data in Fig.S5 is inconsistent between panel a and c.

Reply: We apologize for the mistake. We have corrected the labelling in the revised version.

Reviewers' Comments:

Reviewer #1:

Remarks to the Author:

The revised version of the manuscript shows important improvements. I thank the authors for the detailed replies to my comments. If the experimental points fully convinced me, unfortunately, I cannot say the same for the computational part.

I will report my comments and concerns on the following points:

1) The choice of using pseudopotentials with f orbitals is not the best way to reproduce the complex structure of the Tb₂Pc₃ system since the magnetism of the system is given by the Tb(III) ions. On the other hand, the use of pseudos with explicit f orbitals needs the introduction of on-site Coulomb interactions or the use of local hybrid functionals.

The authors have used DFT+U and for this reason, they should safely use pseudos with explicit f orbitals in bases. In this way, it would be possible to access the whole magnetic structure making it possible to calculate the FM and AFM states (between the Tb ions) and see what are the f magnetic effects on the LUMOs energy splitting, and consequently the J between them. Such data would be of support to the model derived by the authors, too.

2) Computed STM images with explicit f orbitals in the bases should be provided so that they can be compared with the experimental ones as already commented in my previous report.

3) In Fig.S8 should be reported the cutoff value of the charge.

4) It could be of interest to the community the inclusion of the following article on both the YSR states and the different possible orientation effects on them when vanadyl phthalocyanine molecules are adsorbed on Pb(111).

In my opinion, only a compelling response to the comments above will make the paper suitable for publication in Nature Communication.

Reviewer #2:

Remarks to the Author:

In response to my report, authors argue that their observations show new physics because the parameter range is different than in earlier studies. The revised version includes a specific comment to this difference, which I consider important for the placement in regard to earlier literature.

Reviewer #3:

Remarks to the Author:

The authors have properly reply to my concerns. I would therefore recommend the manuscript for publication.

Reply to Review

Reviewer #1 (Remarks to the Author):

The revised version of the manuscript shows important improvements. I thank the authors for the detailed replies to my comments. If the experimental points fully convinced me, unfortunately, I cannot say the same for the computational part.

Thank the reviewer for acknowledging the improvement our manuscript. Our point-by-point responses to the remaining concerns on the DFT part are below.

I will report my comments and concerns on the following points:

- 1) The choice of using pseudopotentials with f orbitals is not the best way to reproduce the complex structure of the Tb₂Pc₃ system since the magnetism of the system is given by the Tb(III) ions. On the other hand, the use of pseudos with explicit f orbitals needs the introduction of on-site Coulomb interactions or the use of local hybrid functionals. The authors have used DFT+U and for this reason, they should safely use pseudos with explicit orbitals in bases. In this way, it would be possible to access the whole magnetic structure making it possible to calculate the FM and AFM states (between the Tb ions) and see what are the f magnetic effects on the LUMOs energy splitting, and consequently the J between them. Such data would be of support to the model derived by the authors, too.

We thank the referee for raising this important point. We agree with the referee that the full f-electron treatment is necessary to evaluate total magnetic moments. However, in our study, we focus on the spin at the Pc ligand state, not the total magnetic moments. We emphasize that using the pseudopotential of the Tb(III) ion with a fixed f-electron state is a standard method for assessing the ligand electronic state of TbPc₂. For example, Tb(III) pseudopotential in VASP was used in a recently published paper in ACS Nano, DOI: 10.1021/acsnano.1c11221. This treatment is rationalized by the strong localization of Tb f-electrons in the phthalocyanine complex, which makes the interaction between f- and π -electrons weak. The small hybridization between f- and ligand π -electrons has been confirmed in the previous research (L. Vitali *et al.*, Nano Lett. 8, 3364–3368 (2008)). Therefore, our calculation reasonably describes HOMO and LUMO (or SOMO) states in TbPc₂ and Tb₂Pc₃. This is further supported by the detailed comparison between ab-initio calculations with fixed f-state and full f-electron pseudopotential (I. I. Vruble *et al.*, Phys. Rev. B 101, 125106 (2020)). In this paper, the density of states (DOS) of TbPc₂ obtained from

two methods was compared, and it was found that the energy difference between HOMO and SOMO is comparable in the two cases.

We conducted a similar comparison in the Tb_2Pc_3 molecule by carrying out an additional calculation using full f-electron pseudopotential. The total magnetic moments of Tb_2Pc_3 including f-electrons becomes $12 \mu_B$, which indicates the ferromagnetic coupling between two Tb(III) ions. This result matches the experimental measurement (K. Kathoh *et al.*, Chem. Rec. 16, 987-1016 (2016)). We compared the DOS projected on the p_z orbital of C atoms in Tb_2Pc_3 obtained from the full f-electron, fixed f-electron pseudopotential, and fixed f-electron pseudopotential with DFT+U. The results are summarized in Fig. R1. The position of HOMO and LUMO is similar in these three calculations because the coupling between the f- and π -electrons is weak. Using DFT+U, the energy difference between HOMO and LUMO becomes closer to that in the full f-electron potential case. This supports that our DFT+U calculation using fixed f-electron pseudopotential is reasonable to describe the HOMO and LUMO electronic state.

Fig. R1 Projected density of state for p_z orbitals of C atoms in Tb_2Pc_3 . Black dashed, red solid, blue dotted lines are the results obtained from the fixed f-electron pseudopotential without DFT+U, that with DFT+U, and full f-electron pseudopotential, respectively. For full f-electron pseudopotential calculation, DFT+U with $U = 5.0$ eV is applied for f-electron states.

We also found that the computation using full f-electron potential is hard to converge. This low stability of self-consistent field calculation hampered us from examining the electronic state of $[\text{Tb}_2\text{Pc}_3]^{-1}$ and $\text{Tb}_2\text{Pc}_3/\text{Pb}(111)$ with an explicit treatment of f-electrons. Nonetheless, we believe that calculation results from fixed f-electron pseudopotential are sufficient to support the experimental results based on the above discussion.

For clarity, we have added Fig. R1 to supplementary information as Fig. S12, cited relevant papers as Refs. 50-52, and added the following discussions to Page 12 of the main text.

“We used pseudopotential of Tb(III) ion in VASP with a fixed f-electron state, which is a standard method for assessing the ligand electronic state of the phthalocyanine complex [50]. This treatment is rationalized by the strong localization of Tb f-electrons in the phthalocyanine complex, which makes the interaction between f- and π -electrons weak [51]. We have carried out additional calculations on Tb_2Pc_3 using full f-electron pseudopotential, fixed f-electron pseudopotential, and fixed f-electron pseudopotential with DFT+U. Those calculations deliver similar electronic states of HOMO and LUMO (Fig. S12), further confirming the validity of using fixed f-electron pseudopotential [52].”

- 2) Computed STM images with explicit f orbitals in the bases should be provided so that they can be compared with the experimental ones as already commented in my previous report.

As discussed in the response to the first comment, there is technical problem in including f-electron explicitly in $\text{TbPc}/\text{Pb}(111)$. The assignment to the configuration, $\text{TbPc}/\text{substrate}$ or $\text{PcTb}/\text{substrate}$, has been discussed in the ACS Nano (DOI: 10.1021/acsnano.1c11221) paper. Similar to our case, $\text{TbPc}/\text{NbSe}_2$ exhibits protrusion in the center of molecule, whereas $\text{PcTb}/\text{NbSe}_2$ does not. This indicates that the protrusion at the center of the TbPc molecule in STM image is generally associated with Tb-up configuration.

For clarity, we have added the following sentence to Page 3 of the Supplementary Information. “It is noted that STM images of TbPc on NbSe_2 substrate display similar protrusion (depression) in the molecular center with Tb facing up (down) [Ref. 50 of main text].”

- 3) In Fig.S8 should be reported the cutoff value of the charge.

We added the information of the isosurface level in the caption of Fig. S8 as follows:

“Isosurface level is set to 1.647×10^{-4} e/Bohr³.”

- 4) It could be of interest to the community the inclusion of the following article on both the YSR

states and the different possible orientation effects on them when vanadyl phthalocyanine molecules are adsorbed on Pb(111).

Thanks for bringing us the attention of the interesting study. We assume the reviewer refers to L. Malavolti *et al.*, Nano Lett. 18, 7955–7961 (2018). In this paper, YSR in VOPc/Pb(111) has been observed by STS. The presence of YSR depends on the direction of VO bond. Moreover, the approach of STM tip causes the evolution of the YSR state of the oxygen-up VOPc, which is attributed to the change of coupling between molecular spin and Pb(111) by the effect of the mechanical force introduced by the tip. The mechanism of the hybridization change induced by the STM tip is similar to our case. We have thus cited the paper as Ref. 33 and added the following sentence to Page 6 of the main text. we add the following explanation in the discussion part.

“Similar effect of the tip force has also been observed on an oxygen-up VOPc molecule, whose interaction with the Pb substrate becomes enhanced with decreasing tip height, inducing quantum phase transition of the associated YSR state [33].”

Reviewer #2 (Remarks to the Author):

In response to my report, authors argue that their observations show new physics because the parameter range is different than in earlier studies. The revised version includes a specific comment to this difference, which I consider important for the placement in regard to earlier literature.

Thank the reviewer for recognizing the importance of our finding and the difference to previous studies.

Reviewer #3 (Remarks to the Author):

The authors have properly reply to my concerns. I would therefore recommend the manuscript for publication.

Thank the reviewer for recommending the publication of our manuscript.

Reviewers' Comments:

Reviewer #1:

Remarks to the Author:

In the revised version, the authors have replied to my concerns related to the implicit use of the f-orbitals. They have presented a list of works that appeared in the literature in support of their choice and they have also presented a plot where the effect of the explicit consideration of the f-orbitals is compared to the case they are not.

First of all, I would like to state that I appreciated the authors' effort but they did not convince me that much. In the R1 plot, the inclusion of the f-orbitals HOMO-LUMO gap is not negligible and I would have liked to see also the plot relative to the inclusion of the U parameter on the f-orbitals, too. Such an inclusion, along with an explicit presence of a strong spin polarization coming from the unpaired electrons in the f-orbital, are expected to have some influence on the α β components of the HOMO and LUMO orbitals, the key point of the paper.

The DFT+U approach is an accurate tool to verify the role and the contribution of the f-orbitals in a system. With the fine-tuning of the U values, the accurate reproduction of main electronic properties can indeed be performed (Journal of Chemical Theory and Computation 2019 15 (11), 5987-5997 DOI: 10.1021/acs.jctc.9b00553, ChemPhysChem 2018, 19, 2947 – 2953

10.1002/cphc.201800489, Journal of Applied Physics 130, 145102 (2021);

<https://doi.org/10.1063/5.0058096>, etc). The authors also claim the conclusions of the following paper I. I. Vrubel et al., Phys. Rev. B 101, 125106 (2020), to support their choice. Interestingly, in its appendix, I see that the use of pseudo vs full-f approaches gives similar BUT different HOMO-SOMO-LUMO energy ladders: -4.6/-4.2/-3.4 eV vs -4.58,-4.57/-3.90,-3.89/-3.00 eV.

I consider these differences matter for the key point of this paper and that a check should be performed.

Moreover, several papers are lately appeared in the literature showing the limits of the picture representing the f-orbitals as completely localized and therefore with a "participation in the valence charge rearrangements nonessential", as the authors state. Since minimal changes can involve the changes in the magnetic properties See for instance Chem. Sci., 2020,11, 2796-2809 and Chem. Sci., 2019,10, 7233-7245).

I agree with the authors that working with f-orbitals in lanthanides is not easy but not impossible.

There are still some symbols that are not correctly encoded in the pdf (see lines 191 and 205, for instance)

For the reasons above, the paper cannot be suited for publication in Nat. Comm. in the present form.

Reply to review

Reviewer 1:

REVIEWER COMMENTS

In the revised version, the authors have replied to my concerns related to the implicit use of the f-orbitals. They have presented a list of works that appeared in the literature in support of their choice and they have also presented a plot where the effect of the explicit consideration of the f orbitals is compared to the case they are not.

First of all, I would like to state that I appreciated the authors' effort but they did not convince me that much. In the R1 plot, the inclusion of the f orbitals HOMO-LUMO gap is not negligible and I would have liked to see also the plot relative to the inclusion of the U parameter on the f orbitals, too. Such an inclusion, along with an explicit presence of a strong spin polarization coming from the unpaired electrons in the f orbital, are expected to have some influence on the alpha 6 beta components of the HOMO and LUMO orbitals, the keep point of the paper.

The DFT+U approach is an accurate tool to verify the role and the contribution of the f orbitals in a system. With the fine-tuning of the U values, the accurate reproduction of main electronic properties can indeed be performed (Journal of Chemical Theory and Computation 2019 15 (11), 5987-5997 DOI: 10.1021/acs.jctc.9b00553, ChemPhysChem 2018, 19, 2947 – 2953 10.1002/cphc.201800489, Journal of Applied Physics 130, 145102 (2021); <https://doi.org/10.1063/5.0058096>, etc). The authors also claim the conclusions of the following paper I. I. Vrabel et al., Phys. Rev. B 101, 125106 (2020), to support their choice. Interestingly, in its appendix, I see that the use of pseudo vs full-f approaches gives similar BUT different HOMO-SOMO-LUMO energy ladders: -4.6/-4.2/-3.4 eV vs -4.58,-4.57/-3.90,-3,89/-3.00 eV.

I consider these differences matter for the key point of this paper and that a check should be performed.

Moreover, several papers are lately appeared in the literature showing the limits of the picture representing the f orbitals as completely localized and therefore with a

“participation in the valence charge rearrangements nonessential”, as the authors state. Since minimal changes can involve the changes in the magnetic properties See for instance Chem. Sci., 2020,11, 2796-2809 and Chem. Sci., 2019,10, 7233-7245).

I agree with the authors that working with f orbitals in lanthanides is not easy but not impossible.

Reply: We thank the referee for the additional comment and bringing to our attention further references. To meet the referee’s request of seeing “also the plot relative to the inclusion of the U parameter on the f orbitals”, we have endeavored to perform calculations using full-f potential plus U on both free Tb₂Pc₃ and its charged form with one additional electron [Tb₂Pc₃]⁻¹. We would like to mention that those calculations are very time-consuming, as is also pointed out by the referee that “working with f orbital in lanthanide is not easy”.

The calculation results for free Tb₂Pc₃ with different U at Tb f-orbitals are summarized in Fig. R1. Evidently, the peak positions in the PDOS of the p_z orbital of C atoms are the same for U=3.0, 4.0, and 5.0 eV, which indicates that the ligand orbital is not sensitive to the U parameters in the Tb f-orbitals. The above calculations suggest the two Tb ions are ferromagnetically coupled, resulting in a total absolute magnetic moment of 12.0 μ_B. We also found that U < 2.0 eV and >6.0 eV results in different magnetic states. Thus, we concluded that U from 3.0 to 5.0 eV is the suitable range to reproduce the experimental measurement reported in the previous study (K. Katoh et al., Chem. Rec. 2016, 16, 987-1016).

In addition to the check of U dependence, we also calculated the electronic state of free Tb₂Pc₃ charged with one additional electron, i.e. [Tb₂Pc₃]⁻¹, in full-f potential. We chose U=5.0 eV as a representative parameter, in line with the results presented above. The calculation results of free [Tb₂Pc₃]⁻¹, whose structure is four-fold symmetric, are summarized in Fig. R2. As shown in Fig. R2a, the ligand SOMO state becomes spin polarized. Even though the positions of HOMO and LUMO+1 states are slightly different from the calculation results without f-orbital, the positions of SOMO, SUMO, and LUMO states are almost the same (Fig. R2b). The sign of the magnetic moment at

the ligand is opposite to that in Tb f-orbitals, which indicates the presence of antiferromagnetic coupling between them (Fig. R2c).

Fig. R1 Calculation of free Tb_2Pc_3 . PDOS of p_z orbital of C atoms with different U parameters of Tb f-orbitals. The energies of the HOMO-1, HOMO, LUMO, and LUMO+1 are the same for $U=3.0, 4.0,$ and 5.0 eV.

Fig. R2 Calculation of free $[\text{Tb}_2\text{Pc}_3]^{-1}$. (a) Calculated DOS projected on the p_z orbitals of C and N atoms in $[\text{Tb}_2\text{Pc}_3]^{-1}$ with $U=5.0$ eV at f-orbital electrons. (b) Comparison between the calculated PDOS of the C- p_z orbitals with/without f-orbital electrons. (c) Top and side view of the calculated spin distribution in $[\text{Tb}_2\text{Pc}_3]^{-1}$ with f-orbital electrons. The majority-spins (minority-spins) are colored in red (blue). The isosurface level is $5.0 \times 10^{-4} \mu_B/\text{Bohr}^3$ for both spins. Note that the assignment of majority- and minority-spins are determined by the magnetization at the ligand SOMO.

The other important point in our interpretation of the experimental results is the different occupations in the split LUMO state in the distorted molecule on Pb(111). This is also confirmed by the calculations with f-electrons for the *distorted* $[\text{Tb}_2\text{Pc}_3]^{-1}$ obtained from the structure relaxation on Pb(111). The results are summarized in Fig. R3. Same as the previous calculation without f-electrons, the originally degenerate LUMO states are split into two after distortion (Figs. R3a,b). One of the states becomes half filled with one additional electron, and the other state is almost empty. This difference in the occupation of the two orbitals results in the two-fold spin distribution shown in Fig. R3c. These results support that two important points are valid even when including f-electrons in the calculation: 1) the formation of magnetic moment in the ligand state by charge transfer from the substrate, and 2) the presence of two energetically close orbitals around the Fermi level, one half-filled and the other almost empty.

Fig. R3 Calculation of free $[\text{Tb}_2\text{Pc}_3]^{-1}$ with distortion. (a) Calculated DOS projected on the p_z orbitals of C and N atoms in distorted $[\text{Tb}_2\text{Pc}_3]^{-1}$. (b) Comparison between the calculated PDOS calculation of the C- p_z orbitals with/without f-orbital electrons in distorted $[\text{Tb}_2\text{Pc}_3]^{-1}$. (c) Top view of the calculated majority- and minority-spin distribution in distorted $[\text{Tb}_2\text{Pc}_3]^{-1}$. The majority-spins (minority-spins) are colored in red (blue). The isosurface level is $2.0 \times 10^{-4} \mu_B/\text{Bohr}^3$ for both spins.

In addition to performing the calculations presented above, we have *experimentally* examined whether the Tb-f orbital impacts the spin-orbital YSR states, the key finding of our study. For that, we have investigated another related molecule Y_2Pc_3 on Pb(111), which has similar molecular structure as Tb_2Pc_3 but significant difference in the f-orbitals. As is shown in Fig. R4, the Y_2Pc_3 molecule indicates identical STM morphology (Fig. R4a) and molecular level energies from the STS spectrum (Fig. R4b). From its low-energy spectrum, the molecule also shows Kondo resonance (Fig. R4c) whose resonance width is comparable to that of Tb_2Pc_3 . More importantly, the Y_2Pc_3 molecule also exhibits the two-pair YSR states (Fig. R4d), the same as the Tb_2Pc_3 . All these observations rigorously prove that the spin-orbital YSR states are indeed from the ligand spin of the Tb_2Pc_3 and Y_2Pc_3 , and the role of the f-orbital is negligible in this case. We note that our experiments demonstrate the exchange interaction between the f-metal ion and the ligand spin is sufficiently small and undetectable at 0.4 K, but do not preclude its existence at even lower temperature.

We can thus conclude, from both the full f-electrons plus U calculation and the additional Y_2Pc_3 experiment, that the spin state of the Tb_2Pc_3 ligand is not affected by the Tb ion. For clarity, we have included the above calculation and experimental results as Figs. S12 and S13, added discussions as Supplementary note 2 and in Pages 11 and 12 of the revised main text.

Fig. R4. Kondo resonance and YSR states of Y_2Pc_3 . (a) STM image ($V = -1.0$ V and $I = 10$ pA) of an imbedded Y_2Pc_3 . (b) Tunneling spectra of Y_2Pc_3 ($V = -1$ V and $I = 100$ pA) and Tb_2Pc_3 ($V = 1$ V and $I = 100$ pA), showing similar molecular levels. The energies of Y_2Pc_3 molecular levels (marked with red dashed lines) slightly shift compared to those of Tb_2Pc_3 due to their different locations imbedded in the Moiré pattern, as is also seen in Fig. 1e of the main text. (c) Tunneling spectrum ($V = -10$ mV and $I = 100$ pA) of the same Y_2Pc_3 under 2 T, showing a Kondo resonance. (d) Tunneling spectrum ($V = -10$ mV and $I = 100$ pA) of Y_2Pc_3 , showing the two-pair YSR states. The dashed lines depict the energies of the superconducting coherence peaks of Pb.

There are still some symbols that are not correctly encoded in the pdf (see lines 191 and 205, for instance)

Reply: We apologize for the mistake. In line 191, the missing symbol should be \ll . In line 205, both missing symbols are \sim . Those missing symbols are caused during the format transformation of our manuscript in the submission system.

For the reasons above, the paper cannot be suited for publication in Nat. Comm. in the present form.

Reply: With the above new calculation and experiment results, we believe the referee's concerns are adequately addressed and hope the current form of our manuscript finally meets the high standard of Nature Communications for publication.

Reviewers' Comments:

Reviewer #1:

Remarks to the Author:

The authors have done lots of efforts to improve their paper by performing further calculations. In a nutshell, the new results confirm that the inclusion of f-orbitals is limited in the neutral system but this is not in the charged one. Indeed, no rigid energy shifts were observed within and between the two spin channels. For this reason, I would have liked to see them being reported in the paper as all the Figures including the results obtained by calculations with the f-orbitals included. The same considerations are valid for the exchange couplings: the authors are in the conditions to calculate them: it would be an extra plus for the paper. Anyway, I am happy about the improvements made by the authors.

Reply to review

Reviewer 1: REVIEWER COMMENTS

The authors have done lots of efforts to improve their paper by performing further calculations. In a nutshell, the new results confirm that the inclusion of f-orbitals is limited in the neutral system but this is not in the charged one. Indeed, no rigid energy shifts were observed within and between the two spin channels. For this reason, I would have liked to see them being reported in the paper as all the Figures including the results obtained by calculations with the f-orbitals included. The same considerations are valid for the exchange couplings: the authors are in the conditions to calculate them: it would be an extra plus for the paper. Anyway, I am happy about the improvements made by the authors.

Reply: We thank the referee for recognizing the improvements of our manuscript and the suggestions. Following the referee's recommendation, we have included all the figures obtained by calculations with full f-orbitals into the supplementary information. We have replaced the original Fig. S12 with three new figures, Figs. S12,S13,S14, and cited those figures in Supplementary Note 2.

We assume the referee is concerned about the exchange coupling between the magnetic moments in Tb ions and the ligand spin. To evaluate the upper limit of the exchange coupling strength, we calculate the energy difference between the ferromagnetic and the antiferromagnetic configuration. The antiferromagnetic configuration is 7.4 meV more stable, indicating the exchange coupling is weak. We add this information to supplementary information in the 2nd paragraph of Page 4 as follows:

“The upper limit of the exchange coupling between the Tb moment and the ligand spin can be estimated from the energy difference between the antiferromagnetic ground state and the ferromagnetic state generated by the constraint on total magnetic moment. The energy difference is 7.4 meV, which indicates the exchange coupling is weak.”